# Effects of CLA, Soybean Oil, and Used Soybean Oil from Fish Friers in Sheep Diets on Milk Lipids and Lamb Tissues

**DOI:** 10.3390/ani15040551

**Published:** 2025-02-14

**Authors:** Isaac De Gasperin-López, Juan Manuel Pinos-Rodríguez, Jorge Genaro Vicente-Martínez, Samuel López-Aguirre, Alejandro Taylor Estrada-Coates, Gustavo Contreras-Hernández

**Affiliations:** Facultad de Medicina Veterinaria y Zootecnia, Universidad Veracruzana, Veracruz 91710, Mexico; idegasperin@uv.mx (I.D.G.-L.); jvicente@uv.mx (J.G.V.-M.); samuellopez@uv.mx (S.L.-A.); aestrada@uv.mx (A.T.E.-C.); guscontreras@uv.mx (G.C.-H.)

**Keywords:** conjugated linoleic acid, omega fatty acids, ewe milk

## Abstract

This study evaluated how the different sources of fatty acids added to ewe diets modify the fatty acid profile in their milk, lamb tissues, and the physicochemical characteristics of the milk. Ewes were fed three different diets: one with conjugated linoleic acid (CLA) and pure soybean oil, another with pure soybean oil, and the third with soybean oil discarded after frying fish. Results showed that the CLA diet reduced the fat and total solids content in the ewes’ milk, while increasing linoleic acid, omega-6, and docosahexaenoic acid (DHA) levels in milk, as well as omega-3 in the lamb brain. These findings suggest that CLA supplementation in the diet of ewes could enhance milk quality, animal health, and productivity in sheep farming.

## 1. Introduction

The isomers of conjugated linoleic acid (CLA) are a group of fatty acids produced during the bacterial biohydrogenation of linoleic acid [1,2]. To date, 56 isomers are known, some with extensively studied nutritional properties for both human and animal health [3,4]. Among these isomers, trans-10, cis-12 CLA is outstanding for reducing the amount of fat in ruminant milk, since it is a potent inhibitor of the de novo synthesis of fatty acids [5], which improves ewe body condition [6]. The isomer cis-9, trans-11 CLA possesses properties that are anti-diabetic, anti-arteriosclerotic, modulators of the immune system, reducers of body weight, and carcinogenesis and mutagenesis inhibitors in animals and humans [7,8,9]. The levels of cis-9, trans11 and trans-10, cis-12 CLA isomers protected with calcium soaps in ewe milk can be increased by dietary inclusion [10].

The addition of omega fatty acids during pregnancy and lactation can modify the productive performance of lambs through fetal programming, which originates changes in metabolism and in the endocrine and nervous system. Moreover, these fats can accumulate in tissues, which improves meat quality, making it healthier for human consumption [11]. While previous studies have explored CLA supplementation in ruminant diets, this research stands out by specifically comparing the effects of different dietary fat isomers during pregnancy and lactation. Furthermore, it goes beyond the typical focus on milk production and includes a detailed analysis of lamb tissues. This is important because the effect of the mother’s diet on a lamb’s brain development is an aspect that is not often emphasized in similar research. The inclusion of soybean oil discarded after frying fish adds an additional layer of novelty, as it investigates a potentially low-cost and sustainable source of dietary fat.

The objective of this study was to evaluate the inclusion of three sources of fatty acids in ewe diet to determine how they are released into milk and later deposited in lamb tissues.

## 2. Materials and Methods

This study was conducted on the ranch “Torreón del Molino” of the Facultad de Medicina Veterinaria y Zootecnia, Universidad Veracruzana, located in the municipality of Veracruz, Mexico, at the coordinates 19°10′11.3″ N; 96.12°10′11.3″ W, altitude 15 m, with an average annual temperature of 26 °C and a relative humidity of 48%. Animals and facilities are property of Universidad Veracruzana.

Forty-five multiparous East Friesian × Katahdin ewes, 32 to 35 months old, 70 days pregnant and with an average liveweight of 42 ± 4 kg were housed in individual pens with dimensions of 2 × 2.5 m with automatic troughs and waterers in sheds with natural ventilation, where they received one of the following integral diets with a daily intake of 3% dry matter/kg body weight during pregnancy and 3.5% during lactation (Table 1): T1 diet with 2% soybean oil and 1% conjugated linoleic acid Lutrell^®^ Pure (BASF, Ludwigshafen, Germany, 10% trans-10 cis-12 and 10% cis-9 trans-11 and 80% stearic acid calcium soap); T2 diet with 3% pure soybean oil; and T3 diet with 3% soybean oil discarded after frying sea fish (*Centropomus undecimalis*, *Pagrus pagrus,* and *Caranx crysos*); the oil used for frying was heated to 180 °C for two hours before being discarded. All diets contained 65% hay (*Digitaria decumbens*) and 35% concentrate. Fats were added to the concentrate during mixing and the fat doses were established based on the dry matter of the feeds. The diets were formulated following NRC [12], and weekly samples of the offered feed were collected and made into a composite sample for chemical analysis (AOAC) [13]. Metabolizable energy was estimated using the procedure described in NRC [12].

Fourteen days postpartum, a sample of 50 mL of milk was collected by manual milking from each ewe to determine the fatty acid profile, the content of total solids, fat, proteins, and lactose by means of ultrasonic generation with the analyzer Lactoscan^®^ (MCC Milkotronic, Nova Zagora, Bulgaria).

The milk and tissues were initially lyophilized and stored in hermetic containers at 20 °C for later processing using the technique described by Ruíz et al. [14] and Berdeaux et al. [15].

The feed samples were processed using the one-step acid methylation technique proposed by Sukhija and Palmquist [16]. The fatty acid profiles of all the samples were determined by a gas chromatograph (HP™ model 6890 GC, Hewlett-Packard™, Sunnyvale, CA, USA) equipped with an HP-Innowax polyethylene glycol column (30 m × 0.316 mm × 0.25 μm; Hewlett-Packard™, Sunnyvale, CA, USA). The pattern of retention times was “PUFA No.1 Marine Source, Analytical Standard” (Sigma-Aldrich™, Darmstadt, Germany).

On the day of weaning (70 days postpartum), 15 lambs selected randomly from each experimental treatment were sacrificed following the Official Mexican Norm (NOM-033-ZOO-1995) “Humanitarian sacrifice of domestic and wild animals” to take samples of perirenal adipose tissue, muscle, and frontal lobe brain tissue.

The data were analyzed under a completely randomized design with three treatments (diet with pure soybean oil and CLA, diet with pure soybean oil, diet with soybean oil discarded after frying sea fish), and 15 ewes per treatment (*n* = 45). The variables were analyzed in a one-way ANOVA model with the GLM procedure of SAS [17]. The treatments were considered fixed components of the model, and the ewes and lambs were random components, where probability values below 0.05 were considered statistically significant.

## 3. Results and Discussion

The isomers of conjugated linoleic acid reduced the quantity of fat and total solids in ewe milk at 14 days of lactation. Protein and lactose did not undergo significant changes due to the type of added fat (Table 2). These results agree with Bauman et al. [6], who reported a reduction of up to 50% in butyric fat in cow’s milk after adding the isomer trans-10, cis-12 CLA to the ruminant diet, and as in our study, the levels of protein and lactose were not affected. 

There are several theories that explain the syndrome of milk fat depression induced by the acid t10, c12-CLA [18,19]. The theory of biohydrogenation is currently the most accepted theory to explain this syndrome. It mentions that the acid t10, c12-CLA is a potent inhibitor of the enzymatic complexes that are responsible for the synthesis, absorption, transfer, desaturation, and formation of triglycerides in the alveolar cells. Specifically, it suppresses the nuclear fragments in protein 1, together with the response of the sterols, which directly regulate butyric fat synthesis, as well as the expression of the thyroid hormone and the receptors activated by peroxisomes [18]. Nevertheless, these are the only mechanisms that have been identified, and so the existence of other physiological components implicated in the milk fat depression syndrome in ruminants caused by the isomers of conjugated linoleic acid has not been ruled out [3].

The fatty acid profile of the experimental diets was modified by the type of added fat (Table 3). The diet to which CLA was added (T1) was that with which a higher percentage of desaturation was obtained. This is because these isomers are protected in calcium soap, which includes 80% stearic acid. CLA isomers were quantified only in the diets that included Lutrell^®^ Pure; it was determined that this product includes the isomers c9 t11-CLA and t10, c12-CLA in similar proportions.

In the diets that included pure soybean oil and soybean oil discarded after frying sea fish, no CLA isomers were quantified because they can only be synthesized naturally by ruminal bacteria [3].

In contrast with what has been observed in non-ruminants, the fatty acid profile in the diet of ruminants does not significantly modify the lipid profile of the milk when using fats and oils with different degrees of saturation without protection [20,21,22].

In this study, we identified and quantified long-chain omega fatty acids in the milk that were not present in the ewe diets (C12:0, C20:4n-6, C20:5n-3, C22:5n-3, and C22:6n-39; Table 4). This can be attributed to de novo synthesis in the mammary gland from their precursors, which were similar in the experimental diets and could have, in turn, been produced by ruminal biohydrogenation [3]. In the same way, the ratio of the n6/n3 fatty acids was similar in the experimental diets so that the action of the enzymes Δ5 and Δ6 desaturases in the mammary gland was not limited for the production of essential fatty acids of the milk. Also, we observed an increase in the content of vaccenic acid (C18:1n7) in T1 because some CLA isomers are precursors [23].

The similarity in the milk lipid profile could be partly explained by the fact that in mammals, especially during the early phase of lactation, body reserves are mobilized for milk fat production, since the energy in the diet is not enough to cover the demands of milk production [20]. Considering that, before the test, all the ewes were subjected to the same conditions of management and feed, we can deduce that a large part of the milk fat was from the adipose tissue reserves of the ewe and fatty acids passed into the milk with a minimum desaturation or elongation. 

The lipid profile of the perirenal and muscle fat of the lambs was not modified by the experimental diets (Table 5 and Table 6). In the same way, no changes in the n6/n3 ratio of these tissues were observed. The most abundant fatty acids were palmitic (C16:0), stearic (C18:0), oleic (C18:1n9), and vaccenic (C18:1n7).

The CLA concentration offered in the ewe diets was not enough to modify the lipid profile of their lambs’ tissues. Moreover, the omega fats from the soybean oil discarded after frying sea fish did not cause a significant difference under the conditions of this study. These results differ from those of Osorio et al. [24], who found significant differences in the concentrations of fatty acids in tissues when they compared two groups of lambs: the first group with natural lactation and the second group with milk substitute high in omega-3 and -6 fatty acids, levels that were lower than in our study. In addition, they mention that, for the lambs’ first weeks of life, the milk fatty acids were 80% of the fat accumulated in meat and fat, while the other 20% of these lipid reserves were synthesized de novo.

The addition of CLA and soybean oil discarded after frying sea fish modified the fatty acid profile of the right frontal lobe of the lambs (Table 7). As in the milk, there was a higher content of stearic acid (C18:0) in the brains of lambs whose mothers ingested CLA; this is due to the calcium soap as the vehicle in the commercial product. The treatment with CLA increased the quantity of linoleic acid in the brain because the CLA isomers are omegas-6 and could be elongated and desaturated [25].

The soybean oil discarded after frying sea fish significantly increased the quantity of eicosapentaenoic (C20:5n-3) and docosahexaenoic (C22:6n-3) acids in lamb brains because when sea fish are fried, the omega-3 fatty acids in their tissues were captured in the soybean oil during frying, increasing the quality of medium-chain omegas-3, which could be elongated and desaturated to form part of the glial cell membrane [11,26].

Fatty acids play a crucial role in neurodevelopmental functions. DHA promotes early synapse formation and the development of new neurons in fetuses and newborn animals [27]. It also regulates gene expression during fetal brain development [28], helps maintain membrane fluidity, and supports synaptic transmission [29].

In mammals, the brain contains approximately 35% of lipids, including polyunsaturated fatty acids, which are poorly synthesized in the brain and must be mobilized from the liver to neural tissues [30]. DHA is an essential component of neuronal cell membranes and plays a critical role in the function of membrane-bound enzymes and receptors [30,31]. DHA is vital for maintaining membrane integrity, supporting synaptic function, and ensuring the proper development of the central nervous system. A deficiency in DHA can impair cerebral functions and lead to irreversible damage resulting in neurological disorders [29].

In our study, we were able to increase DHA levels in the brains of lambs by simply adding two lipid sources to the sheep’s diet. This could enhance their viability by promoting greater brain maturity at birth and during lactation.

Given the significance of these fatty acids, further research is needed to explore the effects of polyunsaturated fatty acids in the diets of ewes and their impact on the neuronal development of lambs. Such studies could also reveal how these nutrients may enhance productive parameters during lamb growth.

## 4. Conclusions

Gestation and lactation in ewes represent a great physiological challenge since a large part of the nutrients of their diet and their body reserves are used to maintain the fetus and later to produce milk. Therefore, innovative nutritional strategies are being developed to improve the health of ewes and their lambs. According to our study, the addition of CLA isomers or discarded soybean oil in ewe diet from pregnancy to lactation is advantageous for the ewe since less energy in the form of milk fat is lost. In addition, increasing the concentration of omega fatty acids in ewe diet to improve omega fatty acids in lamb’s brain could be a feed strategy to increase their productivity and health. Therefore, the inclusion of these fats in diets can be carried out at an industrial level to improve the quality of the products and the parameters in sheep farming.

## Figures and Tables

**Table 1 animals-15-00551-t001:** Ingredients and proximate analysis of experimental diets (dry matter basis).

	Source of Fatty Acids		
**Ingredient**	**T1**	**T2**	**T3**
Pangola grass hay (*Digitaria decumbens*)	65.0	65.0	65.0
Ground corn	19.0	19.0	19.0
Soybean meal 48% crude protein	12	12	12
Vitamin and mineral premix	1.0	1.0	1.0
Soybean oil	2.0	3.0	0.0
Lutrell^®^ Pure (CLA)	1.0	0.0	0.0
Used frying oil	0.0	0.0	3.0
**Proximate analysis**			
Crude protein (%)	12.0	12.1	11.9
Neutral detergent fiber (%)	43.9	44.3	44.5
Acid detergent fiber (%)	23.1	23.5	24.0
Ether extract (%)	6.8	6.7	6.5
Metabolizable energy (Mcal/kg)	1.99	1.97	1.94

T1: CLA + pure soybean oil, *n* = 15; T2: pure soybean oil, *n* = 15; T3: soybean oil discarded after frying sea fish, *n* = 15.

**Table 2 animals-15-00551-t002:** Physicochemical composition of ewe milk (%).

	Source of Fatty Acids			
	T1	T2	T3	SEM
Protein	5.1	5.3	5.5	0.14
Fats	4.9 ^b^	5.6 ^a^	5.8 ^a^	0.03
Total solids	15.1 ^b^	16.1 ^a^	16.7 ^a^	0.08
Lactose	4.2	4.4	4.7	0.23

T1: CLA + pure soybean oil, *n* = 15; T2: pure soybean oil, *n* = 15; T3: soybean oil discarded after frying sea fish, *n* = 15. SEM = Standard error of the mean. ^a,b^ average values with different letters in the same row are different (*p* < 0.05).

**Table 3 animals-15-00551-t003:** Fatty acid profiles in experimental diets (g/100 g of total fatty acids).

	Source of Fatty Acids		
	T1	T2	T3
C14:0	0.09	0.07	0.40
C16:0	10.03	9.18	10.74
C16:1n-9	0.02	0.04	0.03
C16:1n-7	0.14	0.16	0.56
C17:0	0.12	0.10	0.12
C17:1	0.05	0.06	0.04
C18:0	13.76	2.86	3.20
C18:1n-9	31.33	37.77	39.5
C18:1n-7	0.95	1.23	1.22
C18:2n-6	32.55	40.03	36.96
C18:3n-3	5.17	6.86	5.43
c9, t11-CLA	1.87	0.00	0.00
t10, c12-CLA	1.86	0.00	0.00
C18:4n-3	0.55	0.52	0.49
C20:0	0.47	0.46	0.47
C20:1n-9	1.04	0.66	0.84
SFA	24.47	12.67	14.93
MUFA	33.53	39.92	42.19
PUFA	42.00	47.41	42.88
n3	5.72	7.38	5.92
n6	36.28	40.03	36.96
n9	32.39	38.47	40.37
n6/n3	6.34	5.42	6.24

T1: CLA + pure soybean oil, *n* = 15; T2: pure soybean oil, *n* = 15; T3: soybean oil discarded after frying sea fish, *n* = 15. SFA = Saturated fatty acids (∑ C14:0, C16:0, C17:0, C18:0, C20:0), MUFA = monounsaturated fatty acids (∑ C16:1n-9, C16:1n-7, C17:1, C18:1n-9, C18:1n-7, C20:1n-9), and PUFA = polyunsaturated fatty acids (∑ C18:2n-6, C18:3n-3, C9, t11-CLA, t10, C12-CLA, C18:4 n-3). n6/n3 = Ratio of n6/n3 fatty acids in the samples.

**Table 4 animals-15-00551-t004:** Fatty acid profiles of ewe milk (g/100 g of total fatty acids).

	Source of Fatty Acids			
	T1	T2	T3	SEM
C12:0	0.10	0.10	0.08	0.12
C14:0	2.30	2.37	2.12	0.02
C16:0	27.32	28.96	27.84	0.01
C16:1n-9	0.57	0.46	0.72	0.19
C16:1n-7	6.54	6.30	6.44	0.05
C17:0	0.24	0.24	0.35	0.02
C17:1	0.30	0.27	0.42	0.03
C18:0	6.15	6.15	6.46	0.12
C18:1n-9	34.12	36.93	37.70	0.23
C18:1n-7	2.24 ^a^	1.58 ^b^	1.45 ^b^	0.19
C18:2n-6	17.63 ^a^	14.23 ^b^	14.17 ^b^	0.12
C18:3n-3	1.01	1.24	0.9	0.01
C20:1n-9	0.55	0.47	0.68	0.31
C20:4n-6	0.39	0.41	0.37	0.52
C20:5n-3	0.11	0.12	0.08	0.09
C22:5n-3	0.08	0.12	0.02	0.04
C22:6n-3	0.35 ^a^	0.05 ^b^	0.20 ^b^	0.12
SFA	36.11	37.82	36.85	0.31
MUFA	44.32	46.01	47.41	0.32
PUFA	19.57	16.17	15.74	0.21
n3	1.55	1.53	1.20	0.28
n6	18.02 ^a^	14.64 ^b^	14.54 ^b^	0.20
n9	35.24	37.86	39.10	0.16
n6/n3	11.62	9.56	12.11	0.19

T1: CLA + pure soybean oil, *n* = 15; T2: pure soybean oil, *n* = 15; T3: soybean oil discarded after frying sea fish, *n* = 15. SFA = Saturated fatty acids (∑ C12:0, C14:0, C16:0, C17:0, C18:0), MUFA = monounsaturated fatty acids (∑ C16:1n-9, C16:1n-7, C17:1, C18:1n-9, C18:1n-7, C20:1n-9), and PUFA = polyunsaturated fatty acids (∑ C18:2n-6, C18:3n-3, C20:4n-6, C20:5n-3, C22:5n-3, C22:6n-3). n6/n3 = Ratio of n6/n3 fatty acids in the samples. SEM = Standard error of the mean. ^a,b^ average values with different letters in the same row are different (*p* < 0.05).

**Table 5 animals-15-00551-t005:** Fatty acid profiles of the perirenal fat in lambs (g/100 g of total fatty acids).

	Source of Fatty Acids			
	T1	T2	T3	SEM
C12:0	0.54	0.48	0.52	0.03
C14:0	4.70	4.61	4.90	0.19
C16:0	18.53	18.01	18.52	0.31
C16:1n-9	0.54	0.53	0.49	0.01
C16:1n-7	0.97	0.90	1.01	0.03
C17:0	0.51	0.69	0.62	0.02
C17:1	0.13	0.18	0.17	0.01
C18:0	29.76	30.16	29.07	0.59
C18:1n-9	30.93	33.20	31.90	0.59
C18:1n-7	8.41	6.39	7.82	0.30
C18:2n-6	2.67	2.67	2.88	0.07
C18:3n-3	0.53	0.51	0.50	0.01
C18:4n-3	1.05	0.96	0.93	0.06
C20:1n-9	0.02	0.04	0.04	0.00
C20:4n-6	0.31	0.25	0.26	0.01
C20:5n-3	0.14	0.14	0.12	0.01
C22:5n-3	0.02	0.04	0.03	0.00
C22:6n-3	0.24	0.24	0.22	0.01
SFA	54.04	53.95	53.63	0.51
MUFA	41.00	41.24	41.43	0.53
PUFA	4.96	4.81	4.94	0.10
n3	1.98	1.89	1.80	0.07
n6	2.98	2.92	3.14	0.07
n9	31.49	33.77	32.43	0.59
n6/n3	1.51	1.54	1.74	0.07

T1: CLA + pure soybean oil, *n* = 15; T2: pure soybean oil, *n* = 15; T3: soybean oil discarded after frying sea fish, *n* = 15. SFA = Saturated fatty acids (∑ C12:0, C14:0, C16:0, C17:0, C18:0); MUFA = monounsaturated fatty acids (∑ C16:1n-9, C16:1n-7, C17:1, C18:1n-9, C18:1n-7, C20:1n-9, and PUFA = polyunsaturated fatty acids (∑ C18:2n-6, C18:3n-3, C18:4n-3, C20:4n-6, C20:5n-3, C22:5n-3, C22:6n-3). n6/n3 = Ratio of n6/n3 fatty acids in the samples. SEM: Standard error of the mean.

**Table 6 animals-15-00551-t006:** Fatty acid profiles in the muscle of lambs (g/100 g total fatty acids).

	Source of Fatty Acids			
	T1	T2	T3	SEM
C12:0	0.36	0.42	0.35	0.03
C14:0	2.63	3.19	2.44	0.21
C16:0	19.00	20.14	19.56	0.48
C16:1n-9	0.51	0.41	0.58	0.07
C16:1n-7	1.81	1.28	1.39	0.18
C17:0	1.65	1.35	1.76	0.16
C17:1	1.41	1.78	2.35	0.20
C18:0	16.03	17.13	17.06	0.35
C18:1n-9	32.18	30.53	30.68	0.86
C18:1n-7	6.29	6.06	5.76	0.30
C18:2n-6	9.48	10.05	10.64	0.64
C18:3n-3	0.53	0.59	0.62	0.05
C18:4n-3	1.36	1.50	1.55	0.11
C20:1n-9	0.22	0.18	0.17	0.03
C20:4n-6	3.39	3.29	2.92	0.49
C20:5n-3	0.71	0.39	0.59	0.07
C22:5n-3	1.36	0.97	1.05	0.20
C22:6n-3	1.08	0.74	0.53	0.19
SFA	39.67	42.23	41.17	0.74
MUFA	42.42	40.24	40.93	0.81
PUFA	17.91	17.53	17.90	1.19
n3	5.04	4.19	4.34	0.39
n6	12.87	13.34	13.56	1.09
n9	32.91	31.12	31.43	0.86
n6/n3	3.36	3.17	3.12	0.39

T1: CLA + pure soybean oil, *n* = 15; T2: pure soybean oil, *n* = 15; T3: soybean oil discarded after frying sea fish, *n* = 15. SFA = Saturated fatty acids (∑ C12:0, C14:0, C16:0, C17:0, C18:0); MUFA = monounsaturated fatty acids (∑ C16:1n-9, C16:1n-7, C17:1, C18:1n-9, C18:1n-7, C20:1n-9); PUFA = polyunsaturated fatty acids (∑ C18:2n-6, C18:3n-3, C18:4n-3, C20:4n-6, C20:5n-3, C22:5n-3, C22:6n-3), and n6/n3 = ratio of n6/n3 fatty acids in the samples. SEM = Standard error of the mean.

**Table 7 animals-15-00551-t007:** Fatty acid profiles in the right frontal lobe of the lamb brain (g/100 g of total fatty acids).

	Source of Fatty Acids			
	T1	T2	T3	SEM
C14:0	0.88	0.89	0.97	0.03
C16:0	22.19	22.35	23.44	0.22
C16:1n-9	0.49	0.44	0.45	0.03
C16:1n-7	0.64	0.62	0.52	0.04
C17:0	0.87	1.04	1.39	0.11
C17:1	2.27	2.77	2.81	0.15
C18:0	22.72 ^a^	21.53 ^b^	20.87 ^b^	0.24
C18:1n-9	20.44	21.22	20.61	0.19
C18:1n-7	7.21	7.50	7.08	0.08
C18:2n-6	0.78 ^a^	0.70 ^b^	0.58 ^b^	0.05
C18:3n-3	0.27	0.44	0.36	0.09
C18:4n-3	0.30	0.47	0.31	0.08
C20:1n-9	1.65	1.74	1.61	0.06
C20:4n-6	7.97	8.01	7.56	0.15
C20:5n-3	0.13 ^b^	0.17 ^b^	0.38 ^a^	0.05
C22:5n-3	1.56	1.38	1.61	0.09
C22:6n-3	9.63 ^a^	8.73 ^b^	9.45 ^a^	0.23
SFA	46.66	45.81	46.67	0.23
MUFA	32.80	34.29	33.08	0.33
PUFA	20.54	19.90	20.25	0.27
n3	11.79 ^b^	11.19 ^b^	12.11 ^a^	0.24
n6	8.75	8.71	8.14	0.15
n9	22.68	23.40	22.67	0.23
n6/n3	0.76	0.80	0.70	0.02

T1: CLA + pure soybean oil, *n* = 15; T2: pure soybean oil, *n* = 15; T3: soybean oil discarded after frying sea fish, *n* = 15. SFA = Saturated fatty acids (∑ C14:0, C16:0, C17:0, C18:0), MUFA = monounsaturated fatty acids (∑ C16:1n-9, C16:1n-7, C17:1, C18:1n-9, C18:1n-7, C20:1n-9), PUFA = polyunsaturated fatty acids (∑ C18:2n-6, C18:3n-3, C18:4n-3, C20:4n-6, C20:5n-3, C22:5n-3, C22:6n-3). n6/n3 = Ratio of n6/n3 fatty acids in the samples. SEM = Standard error of the mean. ^a,b^ average values with different letters in the same row are different (*p* < 0.05).

## Data Availability

The raw data supporting the conclusions of this article will be made available by the authors on request.

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
