# Peer review of "Effects of CLA, Soybean Oil, and Used Soybean Oil from Fish Friers in Sheep Diets on Milk Lipids and Lamb Tissues"

_animals, 2025, doi:10.3390/ani15040551_

Round 1
Reviewer 1 Report
Comments and Suggestions for Authors
This study reports interesting information on the addition of oil in the diet of sheep, as a novelty the addition of waste oil after frying fish. Small requests for clarification are reported in the attached file.

Author Response
Dear reviewer,
All your comments were attended and yellow colored. We appreciate the time that you spent to improve the manuscript.
Kind Regards,
Juan Pinos-Rodríguez
Corresponding author

Reviewer 2 Report
Comments and Suggestions for Authors
Proper and rational feeding of pregnant sheep not only provides the body with the necessary nutrients and energy, but also promotes the birth and proper development of offspring. From this side, the results of the research in this article, aimed at solving the problem of feeding pregnant sheep using various sources of fat, are relevant. Despite this, there are positions that need to be finalized in the article:
1. The introduction should more clearly indicate the novelty of the research. Has such research been conducted before? How does this research differ from similar studies in this area?
2. In Materials and Methods (lines 65-68), it be indicated whether the fat dosage was set based on the dry matter of the feed or by another calculation?
3. In Table 1, remove "dry matter basis" from the title, or remove these words from the column title.
4. Add % to the title of Table 2, and remove it from the table data
5. Units of measurement should be provided in Tables 3, 4, 6, and 7.
6. What practical use do the authors see for the research results?
7. Of the 26 literature sources used, only 6 are from the last 5 years.
Author Response

(The authors gave the same response as above.)

Round 2
Reviewer 2 Report
Comments and Suggestions for Authors
I believe that all the comments that were made have been eliminated. The authors have made adjustments to the novelty of the research, clarified the details in the Materials and Methods. Corrections were made to the tables, sources of literature for the last 5 years were added, and the practical significance of the work was clarified. I believe that the publication can be accepted in this form.
Author Response
Dear Reviewer:
Your recommendations were attended. Attached you will find the letter response. We appreciate the time that yous have spent to review and improve the manuscript with your comments.
Kind Regards,
Juan Pinos-Rodríguez
Corresponding Author
